# Process Design for Value-Added Products in a Biorefinery Platform from Agro and Forest Industrial Byproducts

**DOI:** 10.3390/polym15020274

**Published:** 2023-01-05

**Authors:** Nicolás M. Clauser, Fernando E. Felissia, María C. Area, María E. Vallejos

**Affiliations:** IMAM, UNaM, CONICET, FCEQYN, Programa de Celulosa y Papel (PROCyP), Félix de Azara 1552, Posadas 3300, Argentina

**Keywords:** process design, value-added products, biorefinery, levulinic acid, polyhydroxybutyrate

## Abstract

Agroforestry wastes are industrial byproducts available locally such as eucalyptus sawdust (EUC) and sugarcane bagasse (SCB). These byproducts can be used as lignocellulosic raw materials to produce high-value products. This study is a techno–economic analysis of four potential scenarios to produce polyhydroxybutyrate (PHB) and levulinic acid (LA) from hemicellulosic sugars by a fermentative pathway in a biomass waste biorefinery. Mass and energy balances were developed, and technical and economic assessments were carried out to obtain gas, char, and tar from residual solids from autohydrolysis treatment. It was determined that microbial culture could be an attractive option for added-value product production. More than 1500 t/year of PHB and 2600 t/year of LA could be obtained by the proposed pathways. Microbial and enzymatic conversion of LA from sugars could significantly improve energy consumption on the conversion strategy. The products from solid residual valorization (char and tar) are the most important for economic performance. Finally, a variation in specific variables could mean substantial improvements in the final indicators of the processes, reaching a higher NPV than USD 17 million.

## 1. Introduction

In the last years, a significant increase in the worldwide consumption of water, food, and energy was reported. It is expected that by 2030, the global demand for water and energy could grow by about 50 and 40%, respectively, while a 35% increase in food production could be required [1].

In this sense, biorefineries offer new options for biomass valorization that can considerably improve the current value chains and replace existing products obtained from fossil sources. Therefore, biorefinery process assessment for their implementation on an industrial scale is crucial for the transition to green industries. In addition, several operations and reactions are involved in biorefinery processes susceptible to optimization.

One of the most relevant steps in biorefinery processes is pretreatment, in which the biomass is fractionated for later recovery. A pretreatment step usually involves the treatment of the biomass at increased temperature and pressure and high liquid-solid ratios, among others. Hydrothermal treatment (also known as liquid hot water treatment or autohydrolysis) is one of the most widely used treatments. It is applied to lignocellulosic materials in diverse operational conditions, e.g., at different temperatures, particle sizes, residence times, and liquid-to-solid biomass ratios (LSR). After hydrothermal treatment, two main streams come from the process: a liquid fraction rich in hemicellulosic sugars and a solid fraction rich in cellulose and lignin, which must be valorized. Several valorization alternatives for different products and raw materials must be evaluated to determine the best schemes for an economically viable biorefinery platform.

One of the most relevant aspects of biorefinery design and analysis is the definition of the products to be produced [2]. In recent years, the production of chemical products, composite materials, pharmaceuticals, biopolymers, food products, and additives, among others, has gained interest.

Microorganisms can produce various valuable chemicals for biotechnological applications (pharmaceutical, nutraceutical, and cosmetics industries) [3]. However, these processes need to achieve techno–economic feasibility, sustainability, and renewability [4,5].

Levulinic acid (LA), formic acid (FA), and furfural production processes via catalytic pathways were developed and analyzed in previous studies. Additionally, energy integration and reuse strategies were proposed to improve economic indicators [6,7]. The obtained economic indicators were attractive. However, these indicators are sensible for several economic and process variables, demonstrating the need for new valuation strategies complemented by previously developed approaches. Recently, fermentation pathways for LA production and microorganism groups capable of converting LA from sugars were reported. For example, the production of LA through the biochemical route has been evaluated recently as an alternative to the catalytic route [8,9,10]. However, no microbial-specific species able to produce LA have been identified, which could be achieved in the short term due to progress in studies of LA microbial synthesis [10,11]. This work assesses the potential of the biochemical route to obtain LA, the fermentative processes involved, the downstream process, and costs.

On the other hand, poly-3-hydroxybutyrate (PHB) is a bio-based biodegradable plastic produced by various microorganisms (*Alcaligenes eutrophus*, *Azotobacter vinelandii*, *Escherichia coli*, others). Its commercialization is limited due to its high production cost compared to conventional petroleum-based polymers. The bioconversion process requires pure cultures and substrates, sterile conditions, solvent extraction, and recovery of PHB. It can be used, among others, in disposable packages, agricultural systems, medicine and medical devices, and sustained drug delivery systems. PHB can be produced through microbial culture by combining various substrates under different growth conditions. The utilization of sustainable resources and waste materials has been established as reasonable and promising for substrates for the production of PHB [12].

This study is a techno–economic analysis of four potential scenarios to produce PHB and LA from hemicellulosic sugars by a fermentative pathway in a biomass waste biorefinery. Hydrothermal treatment was selected as the raw material pretreatment to obtain a rich liquid stream of hemicellulosic sugars, whereas pretreated solid waste was used for char and tar production. Additionally, some process and economic variables were selected to perform a sensitivity analysis for the alternatives, varying one variable at a time (OAT test) to determine how these would influence the economic indicators.

## 2. Materials and Methods

### 2.1. Raw Materials

Eucalyptus sawdust (EUC) and sugarcane bagasse (SCB) were selected as raw materials to compare the alternatives to produce added-value products from the hemicellulosic fraction. Their chemical composition was determined in previous work. The chemical composition of EUC (total on od. wood) was: 41.8% glucans, 10.7% xylans, 1.41% acetyl groups, 32.3% lignin, 7.86% extractives, and 0.59% ashes [13], while for SCB, it was: 43.1% glucans, 23.8% xylans, 1.7% arabinans, 1.7% acetyl groups, 21.3% lignin, 4.8% extractives, and 1.5% ash [14].

### 2.2. Liquid Fraction Valorization

The scheme of the liquid fraction processing for hemicellulose valorization is shown in Figure 1. Two main scenarios were proposed based on the used raw material, EUC, and SCB.

### 2.3. Autohydrolysis

Autohydrolysis pretreatment alternatives were selected from previous studies. Both raw materials were treated with hot water under isothermal conditions. For EUC, conditions were: a temperature of 180 °C and a liquid/solid ratio (LSR) of 6 for 20 min [13]. SCB was subjected to the same time and temperature but with an LSR of 10:1 [14]. After autohydrolysis, spent liquors were concentrated and post-hydrolyzed.

### 2.4. Concentration and Post-Hydrolysis

Hemicelluloses solubilized in the autohydrolysis step were post-hydrolyzed to convert oligomers into monomers. Next, the spent liquor outcoming autohydrolysis process was pumped into an evaporator where liquor was concentrated until 200 g/L of sugars to reduce post-hydrolysis equipment size and operation costs [15]. This step uses a falling film triple effect evaporator, extensively employed in the sugar industry to concentrate sugar solutions [16,17].

Different acids and concentrations can be used for the post-hydrolysis step [2]. In this case, treatments with 0.5 and 4 wt.% H_2_SO_4_ for 60 min at temperatures between 115 and 125 °C were employed for both raw materials, with high conversion yields [14,18]. It was assumed that post-hydrolysis was carried out at 121 °C for 60 min and 1% H_2_SO_4_. After pretreatment steps (autohydrolysis, concentration, post-hydrolysis, and neutralization), the post-hydrolysis spent liquor composition was estimated from previous studies [13,14] (Table 1).

### 2.5. Process Conversion

After post-hydrolysis, the hemicellulosic-sugar-rich spent liquor was sent to a microbial culture step. Microbial culture conditions were selected from a previous study, which evaluated two microbial culture alternatives for the production of levulinic acid (LA) and polyhydroxybutyrate (PHB) from a similar stream [9].

The following alternatives were assumed for both EUC and SCB scenarios because of their similar spent liquor composition after post-hydrolysis (as %):Alternative I: the hemicellulosic fraction was sent to microbial culture. After microbial culture, LA and PHB yields were 31 and 3%, respectively, from the initial carbon source.Alternative II: the hemicellulosic fraction is enriched with acetate to reach 50% the total composition. LA and PHB yields after microbial culture were 31 and 3%, respectively, of the initial carbon source.

After adding the nutrients, the microbial culture proceeded at 25 °C for 40 days of operation. Then, a constant concentration of LA and PHB was observed. For more details, see Pinto-ibieta et al. [9]. Assumptions for the microbial culture step in the technical and economic assessment include energy consumption for cooling and a unit cost for nutrients.

### 2.6. Process Recovery

The outcoming microbial culture stream was filtered to recover the PHB and LA from the cell mass. The recovery processes of PHB from the cellular mass are still under development. The strategies involve solvents or cellular lysis, each with advantages and disadvantages. The recovery using solvents was considered for this study, assuming a unit cost of USD 1 per kg of recovered PHB [19]. After microbial culture, cellular mass was filtered for the solvent extraction step. The selected solvent is CHCl_3_, widely used for PHB [19]. This non-polar solvent allows the extraction of PHB. Since previous studies have demonstrated that LA is not solubilized in non-polar solvents [20], the solvent must be recovered by evaporation after solvent extraction of PHB. After PHB separation, LA was extracted using furfural. Finally, LA and furfural were separated by distillation [21].

The operational conditions for the evaluated processing of the liquid fraction are summarized in Table 2.

### 2.7. Valorization of the Residual Solid

The residual solid of the autohydrolysis is rich in lignin and cellulose (for SCB and EUC), with a moisture content of about 50%. In the last years, thermochemical conversion processes of biomass have gained attention. Products such as syngas, bio-oil, and biochar, among others, could be obtained. These products could be used, i.e., for fuel production, chemicals, and intermediates [23]. Regarding these products, biochar has received increasing attention for several applications in the last few years [24].

One of the most common processes to obtain biochar is pyrolysis, which could be classified into fast and slow pyrolysis [24]. Commonly, fast pyrolysis is used to maximize the bio-oil product yield, while slow pyrolysis is employed to maximize the biochar product yield [25]. Slow pyrolysis, which occurs at a low heating rate and long vapor residence time, targets char as its main product [25]. The usual operational temperatures are between 400 °C and 600 °C, from minutes to hours, with typical yields of about 25–40 wt% char and 20–50 wt% liquid product [24,25,26]. In the present study, pyrolysis was carried out at 450 °C for 60 min, obtaining yields of 35% biochar, 25% bio-oil, 25% gases, and 15% losses. These values are in the range reported in the literature [24,25,27,28].

#### Mass and Energy Balances

The mass and energy balances were conducted considering the main flows involved in each step of the different processes. For the evaporation processes, a steam economy of 2.6 [29], and for microbial culture, an energy requirement of 5 MJ per kg of obtained products [30] is assumed. As previously mentioned, the energy consumption for PHB recovery was taken into account in the cost of this step. The energy consumed is considered as 2 MJ per kg of PHB recovered. This energy is mainly for heating the outcoming liquor from the fermenter and evaporating to separate PHB from the solvent [19]. Finally, the energy for LA recovery is assumed to be 8 kWh per kg of product recovered [7].

The total energy consumption was calculated as the sum of the different energy inputs for the processes presented in Table 2. The utilities, electricity consumption and related equipment, water heating, and cooling were estimated as proposed by Stuart and Halwagi [31].

Regarding the valorization of the residual solid, it was estimated that the energy required to operate the pyrolysis process is between 6 and 15% of the energy available in biomass (about 1.1 to 1.6 MJ/kg) [27,32]. Additionally, it was found that the pyrolysis process could be energy satisfied by the pyrolysis gases generated at temperatures of about 450 °C and, in some cases, at temperatures of about 650 °C [27]. In the present study, it was assumed that the requirement of pyrolysis energy is satisfied by the 100% gases generated in the pyrolysis step.

### 2.8. Economic Assessment

The economic analysis was performed considering the process design. Production costs, labor costs, and capital investment, among others, were estimated [31,33]. Equation (1) was used for the required capital (equipment) estimation based on biomass production at different scales.
(1)$$C=C_{o}{\left(\frac{M}{M_{o}}\right)}^{n}$$

With the previous equation, it is possible to determine the cost (*C*) of a selected capacity (*M*), from reference data (*Co* and *Mo*), and *n* is an exponent smaller than 1 (between 0.6 and 0.8), all adopted for each case from the literature [31,33,34].

Parameters such as scaling factors and installation costs, among others, were estimated from the bibliography [31,33,35,36]. Nutrient costs include an estimation of microorganisms and nutrients added. Labor was determined based on the type of process (batch process) and the facility capacity [35]. Finally, the IRR and the NPV were used as indicators of the profitability of potential investments in biorefinery projects [31,37]. The selected variables for the economic analysis are shown in Table 3.

For pyrolysis equipment, it comprised the pyrolysis reactor, combustor, products recovery, storage, and services [40].

### 2.9. Sensitivity Assessment

For the sensitivity assessment, the final products and byproduct prices, costs associated with raw materials, catalysts, utilities, and the impact of reaction yield variability are the usual analyzed parameters [41]. Additionally, LSR and washing water, among others, are critical process factors [42,43,44]. After the sensitivity variable selection, it is necessary to define the uncertainty limits and the adjusted curves (commonly adjusted curves are normal distribution, uniform, triangular, and lognormal shaped) [41,45]. The normal distribution was the main continuous distribution, and several economic and process factors could be adjusted to it [41,45,46,47].

Several variables could vary according to the adopted value, such as costs of raw materials and energy and prices of products and reagents. Additionally, process variables, such as LSR, sugar concentration, and production scale, among others, could also vary.

A sensitivity assessment was developed for the proposed scenarios to determine the most relevant factors and how their variables affect the economic indicators. Crystal Ball software was used for this analysis. The method is based on the impact of the main uncertain parameters one at a time, keeping the other parameters fixed.

The selected cost variables are raw materials, reagents, energy costs, and product selling prices. Operational variables are LSR, the steam economy of evaporators, available raw materials, and washing water, among others. Therefore, all variables were independently varied by ±20%, assuming a normal distribution around the mean value.

Finally, some technical factors of the first processing steps, assessed in previous studies, were considered to analyze their influence. A sensitivity analysis was developed to determine their impact on the economic indicators.

## 3. Results

### 3.1. Technical Assessment

Based on the adopted considerations, it was determined that for the scenarios analyzed, between 816 and 2685 tons of levulinic acid and between 175 and 1556 tons of PHB could be obtained (Table 4). Additionally, between 8125 and 10,275 t of tar and between 11,375 and 14,385 t of char could be sold or valorized.

Sugar solubilization in the pretreatment step and the total volume of PHB and LA are higher in SCB than in EUC alternatives.

Mass and energy balances show that when using EUC, it is possible to obtain between 4 and 16 kg of PHB and between 21 and 36 kg of LA, whereas for SCB, it is possible to produce between 5 and 31 kg of PHB and between 24 and 54 kg of LA, all on a one-ton base. Figure 2 and Figure 3 present mass and energy balances for both alternatives.

The difference in the obtained products is due to the sugar solubilization in the different scenarios. For SCB, solubilization in pretreatment is higher than EUC; therefore, more LA and PHB could be obtained for the selected alternatives. On the contrary, for EUC scenarios, products from the solid fraction after pretreatment are higher than in the SCB scenario.

Concerning the difference in the production of PHB and LA, in the alternative without acetate, the selected microorganisms probably can synthesize mainly LA and, to a lesser extent, PHB. With the addition of acetate, molecules such as acetic acid can lead to the accumulation of storage polymers such as PHB [9]. PHB seems to have fewer steps to be produced from acetate, which could explain the higher amount of PHB in the alternatives with the addition of acetate.

Regarding the residual solid after autohydrolysis, one of the common uses is as an energy source for the production processes. In the present study, for EUC and SCB alternatives, it is possible to produce 2740 kWh and 2167 kWh, respectively, considering an LHV of 15 MJ/kg and a boiler efficiency of 80%. In both cases, in this sense, the solid residual could be used to cover 80% and 40% direct energy consumption for EUC and SCB pathways, respectively.

This study aims to valorize all the biomass fractions, so the solid fraction after autohydrolysis was used for char and tar production through pyrolysis conversion (Figure 4). For pretreated EUC, it is possible to produce about 288 kg of char, 206 kg of tar, and 206 kg of gases, whereas for pretreated SCB, it is possible to generate about 228 kg of char, 163 kg of tar, and 163 kg of gases. The gases obtained after pyrolysis are used for the energy needed in all involved steps of the pyrolysis process. Based on our assumptions, 913 and 1155 kWh/dry ton of EUC and SCB are needed for drying and pyrolysis processes. Figure 4 shows the scheme of the pyrolysis process assessed.

For energy consumption, the autohydrolysis step and concentration were the higher energy consumers for all scenarios analyzed. In the case of the EUC scenario, energy for autohydrolysis and evaporation represents about 57 and 60% total energy consumption, respectively, whereas in the SCB scenario, it represents about 67 and 71% total energy consumption, respectively. For both scenarios, evaporation is the step with the highest energy consumption (Figure 5). Regarding LA production, in previous studies, energy consumption in the conversion process (acid catalysis) could reach values between 2 kWh/kg of LA and higher values than 3 kWh/kg [7,48]. In the present study, energy could be decreased to low values, about 1.5 kWh/kg (considering the energy for microbial culture and the final LA production); this means an interesting improvement in the energy needed for this product conversion.

### 3.2. Economic Assessment

After mass and energy balances, economic assessments were developed. Initially, the residual solid after pretreatment was evaluated to produce process energy for LA and PHB production, determining that all IRR economic indicators are smaller than 0. For this reason, the results shown for residual solids are those obtained by the pyrolysis process.

For all analyzed scenarios, energy (mainly steam) is one of the most important factors, followed by operational costs. When using acetate, this reagent is one of the most relevant costs. The raw material, water, and acid costs are less influential in the economic balance. Figure 6 presents the results of this analysis.

The economic assessment was carried out for the analyzed scenarios. IRR and NPV were selected as economic indicators, and a discount rate of 15% was selected for the NPV. The highest economic indicators are for EUC scenarios, with an IRR of about 14.4 and 16% and NPV of USD −2.9 and 4.5 million. Indicators for SCB scenarios reach IRR between 5.5 and 7.4%, whereas, with NPV, values are much less than 0 (Table 5).

The investment needed for the proposed scenarios varies between USD 113 and 146 million. The highest investment cost is for SCB scenarios.

For the alternatives with the addition of acetate (alternatives with the higher PHB production) for the EUC and SCB scenarios, the minimum selling price of PHB was determined to obtain positive NPV. Figure 7 presents the selling prices for PHB in the EUC and SCB scenarios.

For the EUC scenario, the selling price of PHB should be increased by about 15%; for the SCB scenario, PHB should be increased up to USD 17/kg.

### 3.3. Sensitivity Assessment

A sensitivity analysis was developed after the economic assessment. The most relevant variables to the economic balance are the price of char and tar for all scenarios. Specifically, for the EUC scenarios, the available raw material is also a key factor, and to a lesser extent, LA and PHB prices and labor and steam cost. For SCB, the available raw material is less important; however, LA and PHB prices have more influence than in EUC scenarios. Results are presented in Figure 8.

### 3.4. Case Study

Finally, some variables were evaluated for EUC scenarios based on data from the Northeast Region of Argentina (NEA) and data already obtained by our research group. Some aspects regarding biomass fractionation were collected in previous studies. Washing water could be a critical factor regarding sugar recovery after pretreatment [49]. The LSR of the pretreatment could influence the sugar solubilization [14] and play an upstanding role in the economic balance of the processes [7].

Regarding the raw material, it is expected that the total waste available in the region could probably decrease in the coming years, so its price could increase because of the use of wood waste for energy generation and the new alternatives for producing high-value-added products [50].

Because of the above, a sensitivity analysis of the proposed schemes was assessed, evaluating selected parameters and their variation in the NPV values. Each variable was varied by 20% from the base case. The accumulative variation of each variable is shown in Table 6.

If some variables are improved, the economic indicators could increase considerably from the base case. In the design of processes, all variables could be evaluated and considered to decrease the uncertainty.

## 4. Discussion

This study shows alternatives to producing value-added products from biomass as byproducts to improve the overall performance of the processes. Recent studies were conducted on polymers developed from several feedstocks [51,52,53]. The selling price of polymers reported in the last years was almost USD 7/kg of PHA, and that of PHB is still higher [53]. Its market prices could be up to 80% higher than petroleum-based polymers [54]. Regarding the production costs, studies have reported productions starting at USD 2.6/kg [51] and higher than USD 20/kg [53,55]. The selling prices determined in the present study are in the range of reported values; additionally, several improvement alternatives could be carried out to decrease the production costs and the final selling prices of the products.

A recent study showed that, in a multiproduct biorefinery, it is possible to produce up to 50 kg per ton of feedstock of PHB from glucose, while lignin fraction is used for electricity [56]. The present study demonstrates that PHB, LA, and value-added co-products can be obtained from all the biomass fractions. Additionally, in the case of LA, as was discussed previously, the biological pathway could significantly improve the energy consumption in the conversion process, allowing an improvement in the technical and economic balances.

## 5. Conclusions

This study assessed PHB and LA production from eucalyptus and sugarcane bagasse hemicellulosic sugars by a fermentative pathway in a biomass biorefinery. Additionally, gas, char, and tar could be obtained from the residual solids of the autohydrolysis treatment.

The mass and energy balances show that microbial culture could be an attractive option to obtain added-value products from EUC and SCB byproducts obtaining more than 1500 t/year of PHB and more than 2600 t/year of LA. Additionally, it could be an attractive alternative to improve energy consumption in process conversion.

The products from the solid fraction valorization (char and tar) are the most critical variables for economic performance. Depending on the feedstock, production could reach more than 10,000 t/year of tar and more than 14,000 t/year of char.

For the assessed scenarios, EUC alternatives are the most attractive options, reaching NPV higher than USD 4 million for the alternative without acetate.

Finally, the simultaneous variation of specific variables could mean substantial improvements in the final indicators of the processes, reaching an NPV higher than USD 17 million.

A result of this study is to highlight that once laboratory scale processes are developed and optimized, technical and economic assessments will be carried out to evaluate the performance at the commercial scale. That will make it possible to determine how the main variables and parameters influence the overall process.

## Figures and Tables

**Figure 1 polymers-15-00274-f001:**
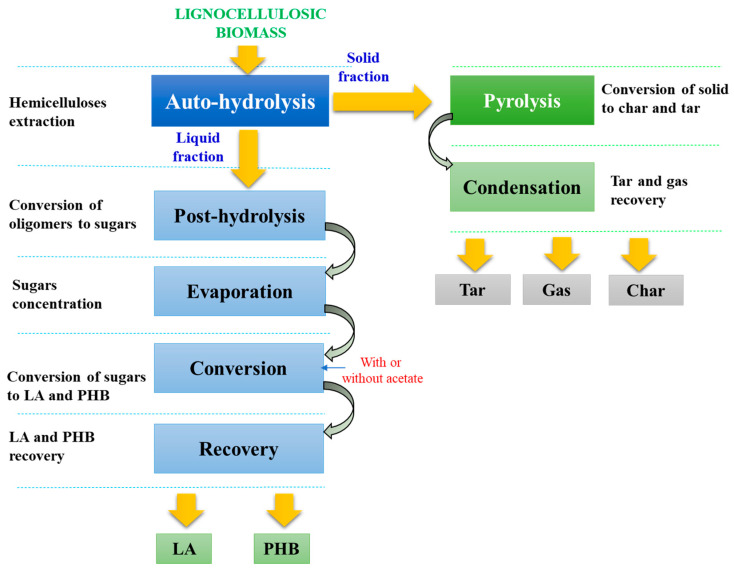
Processing scheme for hemicellulose valorization from the autohydrolysis liquid fraction.

**Figure 2 polymers-15-00274-f002:**
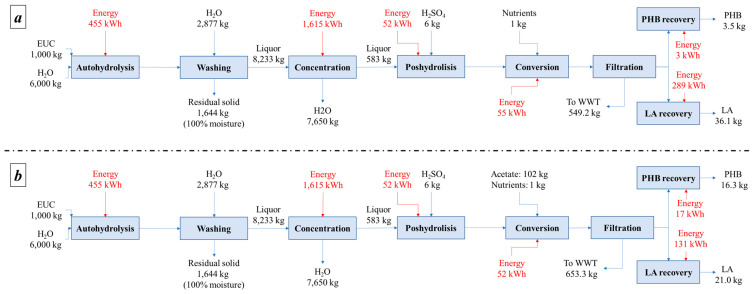
Mass and energy balances for the EUC valorization strategy (**a**) without acetate and (**b**) with acetate.

**Figure 3 polymers-15-00274-f003:**
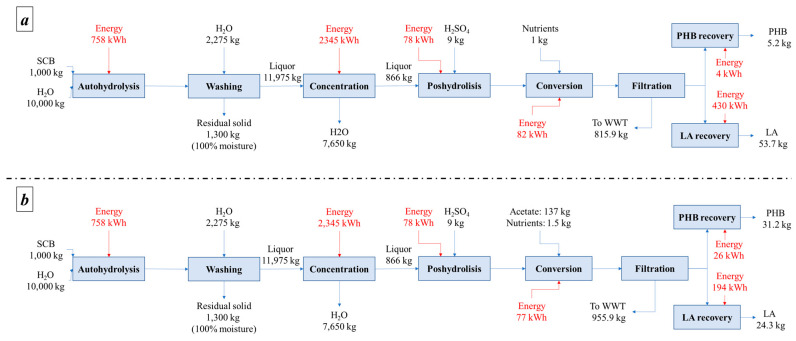
Mass and energy balances for the SCB valorization strategy (**a**) without acetate and (**b**) with acetate.

**Figure 4 polymers-15-00274-f004:**
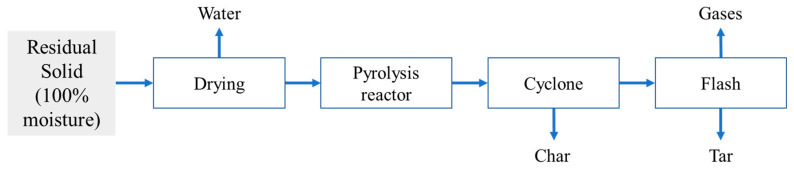
Process flow for solid conversion through the pyrolysis process.

**Figure 5 polymers-15-00274-f005:**
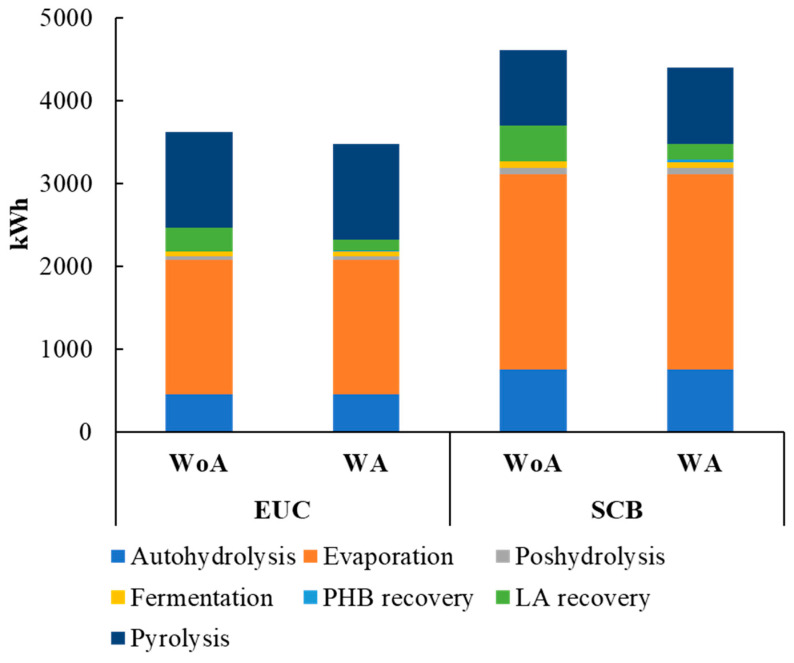
Energy consumption for each assessed scenario is based on dry tons of biomass. WA: with acetate. WoA: without acetate.

**Figure 6 polymers-15-00274-f006:**
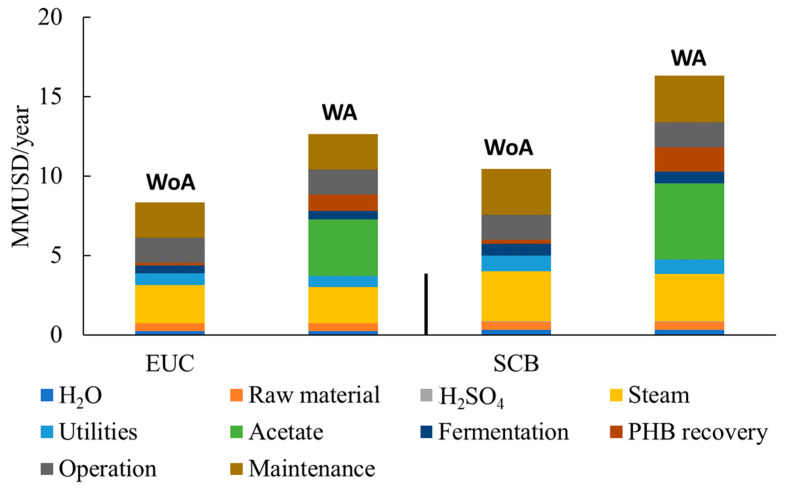
Production costs of selected scenarios. MMUSD: millions of USD. WA: with acetate. WoA: without acetate.

**Figure 7 polymers-15-00274-f007:**
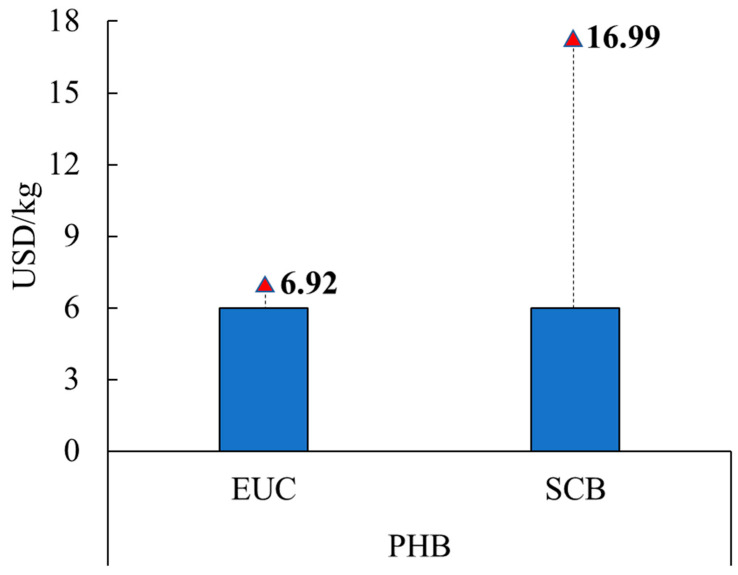
PHB minimum selling price to reach positive NPV. For the scenarios of EUC and SCB with acetate.

**Figure 8 polymers-15-00274-f008:**
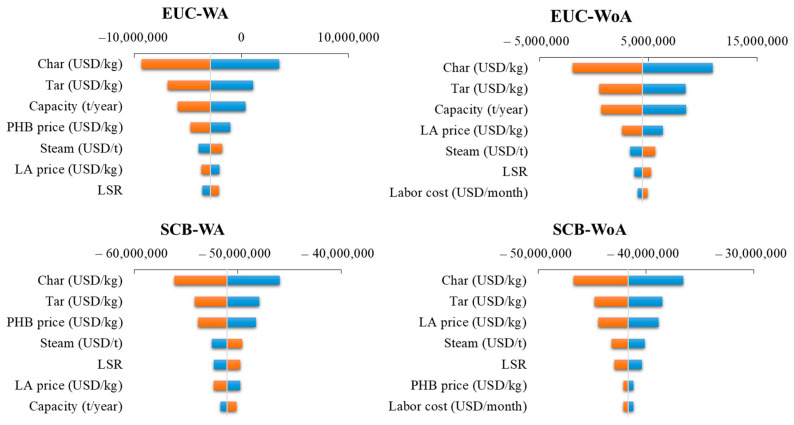
Results of the sensitivity assessment for the selected scenarios. WA: with acetate. WoA: without acetate.

**Table 1 polymers-15-00274-t001:** Composition of spent liquor after post-hydrolysis.

Component	SCB	EUC
	g/L	g/L
Cellobiose	3.40	0.00
Glucose	12.2	15.6
Xylose	147	153
Arabinose	9.10	3.70
Formic acid	1.70	3.10
Acetic acid	20.9	12.6
HMF	0.50	0.20
Furfural	2.80	10.0

**Table 2 polymers-15-00274-t002:** Operational conditions for the evaluated processing of the liquid fraction.

Step	Operational Conditions	Aim	Ref.
Autohydrolysis	EUC: 180 °C, LSR of 6, for 20 min.	Extract hemicelluloses	[13]
	SCB: 180 °C, LSR of 10, for 20 min.	Extract hemicelluloses	[14]
Washing of the solid fraction	3.5 tons of water per ton of dry pulp	Recover all hemicellulosic sugars.	[22]
Evaporation 1	With a steam economy of 2.6	To reach 200 g/L of sugars	[15]
Post-hydrolysis	120 °C for 60 min and 1% H_2_SO_4_	To convert oligomers into monomers	
Conversion	Mixed microbial cultures and nutrients were added to the microbial culture at 60 rpm and maintained at 25 °C for 40 days. *	To convert sugars in PHB and LA	[9]
Filtration	After conversion to recover the cellular mass	To recover cellular mass	
PHB extraction	Using solvent CHCl_3_ at 60 °C.	To recover PHB	[19]
LA extraction	Using furfural as solvent.	To recover LA	[21]

* For more details, see Pinto-ibieta et al. [9].

**Table 3 polymers-15-00274-t003:** Unit values of variables used in the techno–economic assessment.

**Unit Prices at the Mill Gate**		
Sawdust (t/year)	50,000 (Dry basis)	Assumption
Sawdust (USD/t) *^a^*	7.0	
Water (USD/m) *^b^*	0.7	
Electricity (USD/MWh) *^c^*	85	
Labor (USD/h) *^d^*	9–21	
Steam (USD/t)	10	Assumption
Maintenance and taxes	2% (of TCI)	Assumption
Tax rate	35%	Assumption
**Chemicals for production**		
H_2_SO_4_ (USD/kg) *^e^*	0.09	
Nutrients (USD/kg)	10	Assumption
Waste treatment (USD/m^3^) *^f^*	1	
**Products (assumptions)**		
Char (USD/t) *^g^*	1.5	
Tar (USD/t) *^g^*	1.3	
PHB (USD/kg) *^g^*	6	
LA (USD/kg) *^g^*	3.5	

*^a^* Price estimated from the Instituto Nacional de Tecnología Agroindustrial (INTA) [38]. *^b^* Average price in Argentina. *^c^* Energy cost in Misiones, Argentina. *^d^* Value depends on the worker’s position. *^e^* [39]. *^f^* [2]. *^g^* Prices surveyed from international sellers. TCI: total capital investment.

**Table 4 polymers-15-00274-t004:** Total products obtained by each evaluated scenario.

Product	Raw Material			
	EUC		SCB	
	Without acetate	With acetate	Without acetate	With acetate
LA (t/year)	1807	816	2685	1213
PHB (t/year)	175	1049	260	1559
Tar (t/year)	10,275	10,275	8125	8125
Char (t/year)	14,385	14,385	11,375	11,375

**Table 5 polymers-15-00274-t005:** Economic indicators for scenarios evaluated.

	EUC		SCB	
	Without Acetate	With Acetate	Without Acetate	With Acetate
IRR	16.0%	14.4%	7.4%	5.5%
NPV (MMUSD)	4.5	−2.9	NV	NV
Investment (MMUSD)	113	113	146	146

NV: negative value. MMUSD: millions of USD.

**Table 6 polymers-15-00274-t006:** The sensitivity assessment for the selected variables in the EUC scenario. Values of NPV in USD.

		Without Acetate	With Acetate
Increase 20%	Raw material (t/year)	17,610,627	8,654,484
Reduce 20%	LSR	8,604,932	1,141,480
Reduce 20%	Washing water	7,294,834	−168,618
Reduce 20%	Steam	6,793,459	−669,993
	**Base case (USD)**	**4,462,196**	**−2,886,175**
Increase 20%	Steam	2,130,932	−5,102,358
Increase 20%	Washing water	1,458,368	−5,774,923
Increase 20%	LSR	−327,790	−7,561,081
Reduce 20%	Raw material (t/year)	−6,722,415	−12,509,047

## Data Availability

Not applicable.

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
