# Peer review of "Process Design for Value-Added Products in a Biorefinery Platform from Agro and Forest Industrial Byproducts"

_polymers, 2023, doi:10.3390/polym15020274_

Round 1
Reviewer 1 Report
Authors have done nice work with good design for the value added products from agro and forest related byproducts. The only issue is the reference citation style should be revised, and more relvent and recent articles regarding the biorefinery should be included in the manuscript. Overall, a minor revision is suggested.
Author Response
Reviewer 1
Authors have done nice work with good design for the value-added products from agro and forest related byproducts. The only issue is the reference citation style should be revised, and more relevant and recent articles regarding the biorefinery should be included in the manuscript. Overall, a minor revision is suggested.
Response: The following references were added to the manuscript:
Geng, W.; Venditti, R.A.; Pawlak, J.J.; de Assis, T.; Gonzalez, R.W.; Phillips, R.B.; Chang, H. Techno‐economic Analysis of Hemicellulose Extraction from Different Types of Lignocellulosic Feedstocks and Strategies for Cost Optimization. Biofuels, Bioproducts and Biorefining 2020, 14, 225–241, doi:10.1002/bbb.2054.
Ruiz, H.A.; Galbe, M.; Garrote, G.; Ramirez-Gutierrez, D.M.; Ximenes, E.; Sun, S.-N.; Lachos-Perez, D.; Rodríguez-Jasso, R.M.; Sun, R.-C.; Yang, B.; et al. Severity Factor Kinetic Model as a Strategic Parameter of Hydrothermal Processing (Steam Explosion and Liquid Hot Water) for Biomass Fractionation under Biorefinery Concept. Bioresour. Technol. 2021, 342, 125961, doi:10.1016/j.biortech.2021.125961.

Reviewer 2 Report
This study is a techno-economic analysis of four potential scenarios to produce PHB and levulinic acid (LA) from hemicellulosic sugars by a fermentative pathway in a biomass waste biorefinery. Mass and energy balances were developed, and technical and economic assessments were carried out to obtain gas, char, and tar from residual solids from autohydrolysis treatment. The topic of this paper involves the frontier of the discipline, which has important research significance and a wide range of application prospects. In general, the work is well done, while the conclusion is supported by the experimental and results. However, there are still some issues to be addressed before its acceptance.
1. The manuscript should be carefully rechecked to correct the typos and grammar issues.
2. Why the productivity of LA the same while different for PHB with the two methods? Please explain.
3. To present more contents in value-added products and biorefinery by products, some highly relevant, important and recent articles should be included in the manuscript: Integrated lignocellulosic biorefinery: Gateway for production of second generation ethanol and value added products; Second generation biorefining in Ecuador: Circular bioeconomy, zero waste technology, environment and sustainable development: The nexus; Opportunities for New Biorefinery Products from Ethiopian Ginning Industry By-products: Current Status and Prospects; etc.
4. Please compare the advantages and disadvantages of the strategies with previous work.
5. Authors listed the references with numbers. However, the references are cited in the main text with author name. It is suggested to cite the references also in numbers.
Author Response
We greatly appreciate your comments about our manuscript. The responses are in the attached file.
